

# A fresh look at an old concept: home-range estimation in a tidy world

Johannes Signer[1] and John R. Fieberg[2]

[1] Wildlife Sciences, Faculty of Forestry and Forest Ecology, University of Goettingen, Göttingen, Germany
[2] Department of Fisheries, Wildlife, and Conservation Biology, University of Minnesota, St. Paul, MN, USA

## ABSTRACT

A rich set of statistical techniques has been developed over the last several decades to estimate the spatial extent of animal home ranges from telemetry data, and new methods to estimate home ranges continue to be developed. Here we investigate home-range estimation from a computational point of view and aim to provide a general framework for computing home ranges, independent of specific estimators. We show how such a workflow can help to make home-range estimation easier and more intuitive, and we provide a series of examples illustrating how different estimators can be compared easily. This allows one to perform a sensitivity analysis to determine the degree to which the choice of estimator influences qualitative and quantitative conclusions. By providing a standardized implementation of home-range estimators, we hope to equip researchers with the tools needed to explore how estimator choice influences answers to biologically meaningful questions.

# INTRODUCTION

The biological concept of an animal's *home range* has served as a useful construct for organizing our thinking about how animals use and interact with space since the time of Darwin (*Horne et al., 2020*; *Kie et al., 2010*). Today, most people associate the term home range with *Burt*'s (*1943*) definition, "that area traversed by the individual in its normal activities of food gathering, mating and caring for young. Occasional sallies outside the area, perhaps exploratory in nature, should not be considered as in part of the home range." A variety of approaches have been developed to quantify the spatial extent and intensity of landscape use by individual animals and to gain insights into factors that structure their home ranges (see e.g., *Powell, 2012* and associated papers in a special feature on the topic). Recently, *Fleming et al. (2016)* and *Horne et al. (2020)* have argued for classifying statistical home-range methods according to whether they estimate either one of two estimation targets: the *range distribution* or the *occurrence distribution*. The range distribution, sometimes also referred to as the long-term equilibrium distribution, results from an animal continuing to move in a consistent manner, while an occurrence distribution captures an animal's movement path and its associated uncertainty during a specific observation window. This dichotomy is appealing from a theoretical point of view,

Corresponding author
Johannes Signer,
jsigner@uni-goettingen.de

and several new statistical estimators have been developed for targeting these quantities while also addressing issues related to autocorrelation, a prominent feature of current Global Positioning System (GPS) data (*Fleming et al., 2014*; *Fleming et al., 2015*; *Fleming et al., 2016*). These new methods require fitting a movement model to each animal's location data, which is then used to quantify the degree of autocorrelation in the data and also allows one to incorporate location errors (*Fleming et al., 2020*). When estimating the range distribution, the degree of autocorrelation influences the effective sample size used to determine an appropriate smoothing parameter in an autoocrrelated Kernel Density Estimator (aKDE; *Fleming et al., 2015*). When estimating the occurrence distribution, the degree of autocorrelation is used to improve prediction via time-series kriging (*Fleming et al., 2016*).

Despite these advances, many biologists continue to use a variety of estimators (e.g., minimum convex polygons [MCP]; *Mohr, 1947*; or kernel density estimators [KDE] that assume independent location data; *Worton, 1989*) without explicit discussion of a particular estimation target (e.g., *Froy et al., 2018*; *Ranc et al., 2020*). The reasons may include (1) lack of familiarity with recent literature on home-range estimators, (2) lack of confidence in using current estimators that account for autocorrelation but require an extra time- and computational-resource-demanding step, (3) the feeling that this step is not necessary, or (4) interest in estimating something other than a range or occurrence distribution. The new methods developed by Fleming and co-authors are major contributions to this area of research, and these authors have attempted to make these methods accessible to biologists by providing open-source software and training for implementing their estimators (*Calabrese et al., 2020*; *Fleming & Calabrese, 2020*). We suspect, however, that many biologists continue to view traditional home-range estimators as convenient, though imperfect, indices that capture the spatial extent of the area used by individuals during specific tracking periods. For convenience, and as is common in the literature, we will refer to the suite of methods used in this context as *home-range estimators*, even though these methods may have different statistical estimation targets (*Horne et al., 2020*).

Whereas there are many studies that compare different methods for quantifying space use with the goal of determining a single "best" estimator (e.g., *Lichti & Swihart, 2011*; *Noonan et al., 2019*; *Walter, Onorato & Fischer, 2015*), we aim to accommodate multiple estimators in a single analysis to determine whether results are robust to the choice of home-range estimator. As we have argued previously, we think researchers should carefully consider estimators and their properties (e.g., variance or statistical power), and choose one or more depending on the specific biological questions of interest (*Fieberg & Börger, 2012*; *Signer et al., 2015*).

To accomplish our goals, we suggest a general and consistent computational framework for home-range estimation that should be able to accommodate *most* home-range estimators. We propose two classes of home-range estimators (geometric and probabilistic) and a set of properties for each class. Having a standardized representation of home-range estimators facilitates their computation, visualization, and comparisons among estimators. This proposal goes hand in hand with calls for more reproducible and

standardized workflows in (wildlife) ecology (*Archmiller et al., 2020*; *Gula & Theuerkauf, 2013*; *Lewis, VanderWal & Fifield, 2018*). After introducing the framework conceptually, we demonstrate how to estimate home-ranges using this framework following the principles of tidy data (*Wickham, 2014*) using the R package amt (*Signer, Fieberg & Avgar, 2019*) for the R programming language (*R Core Team, 2020*) and a previously published data set on space use by fishers (*Pekania pennanti*) in New York State, USA (*LaPoint et al., 2013a*).

## A CONCEPTUAL FRAMEWORK FOR HOME RANGES AND THEIR ESTIMATION USING amt

Home-range estimators can be divided into two classes: geometric and probabilistic estimators (Fig. 1; *Fleming et al., 2015*). Geometric estimators are constructed following a set of rules and are often hull-based, i.e., the home range is a polygon that is constructed using (all) points where an animal was observed. Typical examples of geometric estimators are minimum convex polygons (*Mohr, 1947*) or local convex hulls (LoCoH; *Getz & Wilmers, 2004*). On the other hand, probabilistic estimators have an underlying probabilistic model and estimate a utilization distribution, the two-dimensional relative frequency distribution of an animal's spatial locations (*Van Winkle, 1975*), from which a hull-based home range can be retrieved for a given isopleth level. Typical examples of probabilistic home-range estimators include uniform or bivariate normal models (*Horne & Garton, 2006*; *Van Winkle, 1975*), traditional KDEs (*Fieberg, 2007*; *Worton, 1989*), and autocorrelated KDEs (aKDE; *Fleming et al., 2015*).

Each home-range estimator, regardless of its class, has several attributes (values stored within the object) and methods (functions to work with the estimate). Estimators of both classes should have the following three attributes: the coordinate reference system (CRS), the data that were used to construct the home-range estimate (data), and the home-range isopleths or levels (levels). The CRS is inherited from the data used to estimate the home range and will be needed to ensure that the home range is correctly positioned in space and that the units of the home-range area are correct. The attribute data contains the original data used to calculate the home range which can be especially useful for plotting home ranges. Finally, home-range areas are calculated for a prespecified home-range level (or isopleth). For probabilistic estimators, the cumulative distribution function of the utilization distribution is truncated at given quantile, with associated $1 - \alpha$ level. For hull-based methods, the outermost points are excluded (given that it is possible to identify the outermost points). It is common to use the 95% isopleth to determine the home range (although arguments for 90% levels exist e.g., *Börger et al., 2006a*). Further, we propose the following methods for all home-range estimators: hr_area() to calculate the home-range area, hr_isopleth() to calculate the home-range isopleths at the specified levels, hr_overlap() to calculate the home-range overlap between two or more home ranges, and plot() to plot the home range.

In addition to the four global methods, specific home-range estimation methods or classes can have additional properties and methods. Probabilistic home ranges, for example, have a property that gives information about the spatial extent and

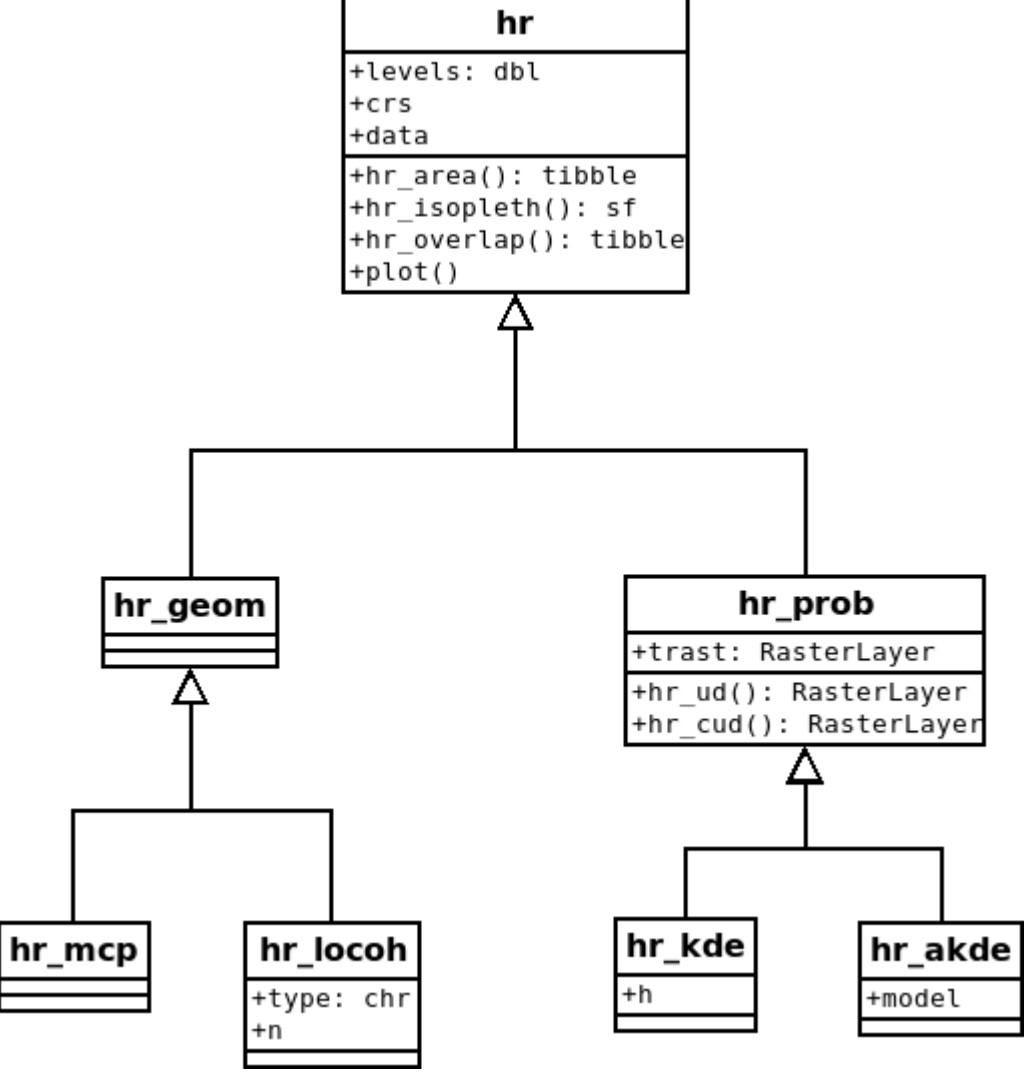

**Figure 1** **Proposed class diagram for home-range estimators.** All home-range estimators will have common attributes (levels, crs [coordinate reference system] and data) and common global methods (hr area() [to obtain the home-range area], hr isopleth() [to obtain the home-range isopleths at the specified levels], hr overlap() [to calculate home-range overlap], and plot()). Probabilistic estimators will also have a common attribute, trast (a template raster for the utilization distribution [UD]), and several additional common methods (hr ud() [to obtain the UD], hr cud() [to obtain a cumulative UD]). Lastly, each individual estimator can have additional properties or methods (e.g., model for hr akde()).

resolution of the utilization distribution (this is termed template raster, or `trast` for the argument name). Probabilistic home ranges have a method to obtain the utilization distribution (`hr_ud()`), the cumulative utilization distribution (`hr_cud()`), and to quantify volumetric intersections of two utilization distributions (implemented within the function `hr_overlap()`). Examples of estimator-specific properties are the number of neighbors used for the local convex hull method, the bandwidth for kernel density estimation, or the movement model used for autocorrelated kernel density estimation (Fig. 1).
An established set of methods makes it easy to work with home ranges. For example, the function hr_area() in the package amt, returns a tibble (*Müller & Wickham, 2020*) with three columns (the home-range level, the home-range area, and an identifier for whether the area is an estimate or a confidence limit), regardless of the estimator. A tibble is an extension of a data.frame in R (i.e., a two-dimensional data structure) but with improved printing and subsetting functions. In addition, a tibble facilitates working with list-columns, which we will demonstrate later in this paper when analyzing data from multiple animals or sampling instances. The function hr_isopleth() in amt returns an sf object (*Pebesma, 2018*). An sf object represents simple features (e.g., points, lines or polygons) in a data.frame with a geometry list-column. The sf (simple feature) package is the successor for the sp package (*Pebesma & Bivand, 2005*) in R to work with spatial data. Having isopleths in an sf-compatible format enables further GIS-related work with the sf package.

## ONE INDIVIDUAL OR SAMPLING INSTANCE

In the first set of examples, we demonstrate how home ranges and derived quantities can be calculated for a single individual or sampling instance (e.g., a home range for one individual using data collected during a single tracking period). We use a data set containing locations of fishers from New York State, USA (*Kranstauber et al., 2011*; *LaPoint et al., 2013a*). These data are freely available from Movebank (*LaPoint et al., 2013b*), and include observations of six individuals (three males and three females) tracked between January and March 2011, with a sampling rate of 10 min or less. We use a preprocessed data set here, where we subsampled the data to form hourly observations. All steps to prepare the data set are provided in https://zenodo.org/record/3991482. For the first few examples, we will use data from one female (F1); for the second set of examples, we will use data from all six individuals.

First, we load required packages, including amt for calculating home ranges (*Signer, Fieberg & Avgar, 2019*), dplyr for data manipulation (*Wickham et al., 2020*), and readr for reading the preprocessed data (*Wickham & Hester, 2020*). For the second set of examples we will need the packages tidyr (*Wickham & Henry, 2020*) to create list-columns and purrr (*Henry & Wickham, 2020*) to iterate over list-columns. After loading the data, we use the function make_track() to create a track –a class used by amt. Within the function make_track() we have to provide columns with the coordinates (x_ and y_), the time stamp (t_), columns with additional variables (here sex, id and HDOP for the location error), and the CRS. A convenient way to specify the CRS is the ESPG (European Petrol Survey Group) code as we have done here (crs = 5070). 5070 is the EPSG code for a projected CRS for the USA (NAD83). We then use filter() from the dplyr package to filter only those relocations that belong to the fisher where id == "F1".

```
library(amt)
library(readr)
library(dplyr)
library(tidyr)
```

```
library(purrr)
dat <- read_rds("data/fisher_preprocessed1.rds") %>%
  mutate(sex = substr(id, 1, 1)) %>%
  make_track(x_, y_, t_, id = id, sex = sex, HDOP = HDOP,
             crs = 5070)
fisher.f1 <- dat %>% filter(id == "F1")
```

With the `fisher.f1` data set, we can now calculate different home-ranges estimates. We demonstrate by calculating MCP and KDE home ranges here (with the default bandwidth, which is chosen to minimize the integrated mean squared error under the assumption the data are normally distributed; *Worton, 1989*). For both home-range estimators, we estimate home ranges at two different home-range levels (50% and 95%).

```
mcp1 <- hr_mcp(fisher.f1, levels = c(0.5, 0.95))
kde1 <- hr_kde(fisher.f1, levels = c(0.5, 0.95))
```

Results from applying any estimator in `amt` are stored in a named `list`. All estimators have three entries in common: `crs`, `data` and `levels`. `crs` stores the coordinate reference system of the home range estimate, inherited from the data used to estimate the home range. The attribute `data` contains the track that was used to estimate the home range (a `track_xy*` from package amt), unless during estimation the argument `keep.data` was set to `FALSE`, then the attribute `data` is NULL. Finally, the argument `levels` contains the home-range levels that were used when estimating the home range.

All estimators also have at least four generic functions for working with the results and for basic plotting. The `plot()` function plots the home range isopleths with the observed points unless `keep.data` is set to `FALSE` or the argument `add.points` is set to `FALSE`. Below, we plot KDE- and MCP-based home ranges. When plotting the MCP, we use the arguments `add.relocations = FALSE` to avoid plotting the observed locations twice; further, we set the argument `add = TRUE` to draw the MCP home range to the existing plot and `lty = 2` to distinguish the KDE home range from the MCP home range by a dashed line type (Fig. 2).

```
plot(kde1)
plot(mcp1, add.relocations = FALSE, add = TRUE, lty = 2)
```

Furthermore, we can now continue to work with these home-range estimates. For example, we can query the home-range area with the function `hr_area()`, which returns a `tibble` with three columns: the home-range level, the home-range area and an identifier for whether the area is an estimate or a confidence limit. Note, at the moment only aKDE returns confidence intervals for home-range area.

```
hr_area(mcp1)
```

```
## # A tibble: 2 x 3
##    level what           area
```
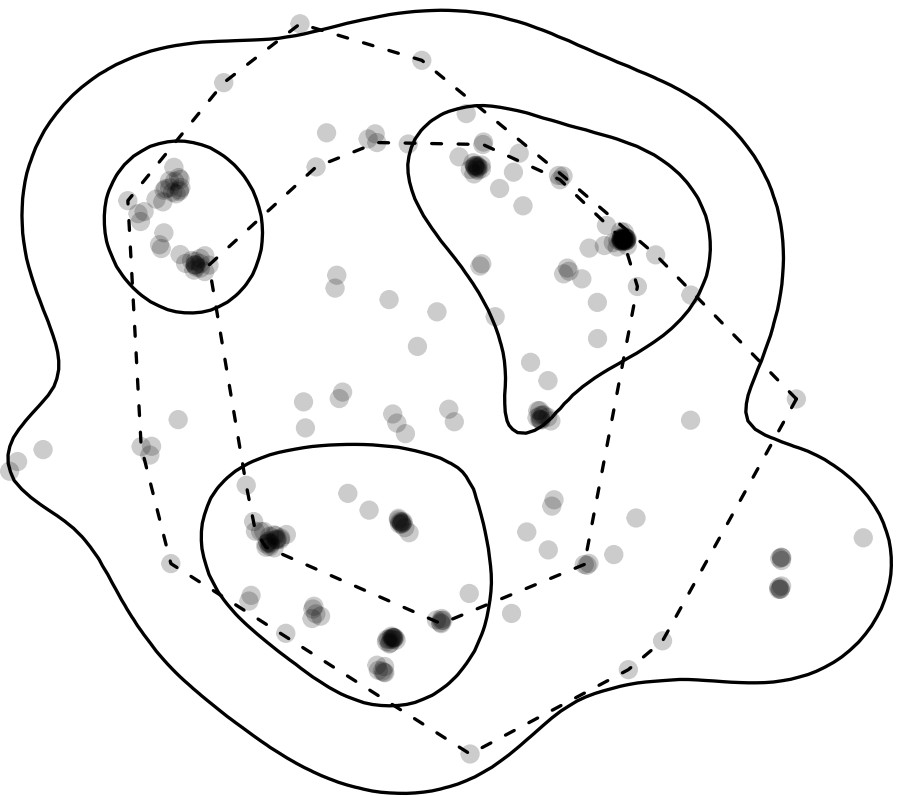

**Figure 2** **Home-range estimates from one fisher.** Points where a fisher was tracked (gray points) over-laid with kernel density (solid lines) and minimum convex polygon (dashed lines) home ranges at two levels (50% and 95%). Note, that whereas home ranges delineated using the 95% level are relatively similar, the home ranges at the 50% level are very different.

```
##   <dbl> <chr>          <dbl>
## 1  0.5  estimate 2298523.
## 2  0.95 estimate 4614598.
```

**hr_area**(kde1)

```
## # A tibble: 2 x 3
##   level what           area
##   <dbl> <chr>         <dbl>
## 1  0.5  estimate 2060451.
## 2  0.95 estimate 7296172.
```

The function hr_isopleth() returns a data.frame with a simple feature column where the home-range isopleths are stored in an sf-compatible format (*Pebesma, 2018*), which we can use to perform further spatial analyses, visually inspect the home range, or export it to a GIS.

```
hr_isopleths(mcp1)
```

```
## Simple feature collection with 2 features and 3 fields
## geometry type:  POLYGON
## dimension:      XY
## bbox:           xmin: 1783582 ymin: 2407247 xmax: 1786138 ymax: 2410039
## CRS:            +init=epsg:5070 +proj=aea +lat_1=29.5 +lat_2=45.5 +lat_0=23
##   level    what          area                          geometry
## 1  0.50 estimate 2298523 [m^2] POLYGON ((1785327 2407972, ...
## 2  0.95 estimate 4614598 [m^2] POLYGON ((1786138 2408604, ...
```

Finally, we may want to calculate the extent of overlap between two or more home ranges (*Fieberg & Kochanny, 2005*), for which we provide the function hr_overlap(). This function can be used to calculate overlap between any two sampling instances (e.g., time periods, animals or even estimators). Below we calculate overlap of the MCP and KDE home ranges, which were previously estimated.

```
hr_overlap(mcp1, kde1)
```

```
## # A tibble: 2 x 2
##   levels overlap
##    <dbl>   <dbl>
## 1    0.5   0.454
## 2   0.95   0.992
```

hr_overlap() by default always calculates the fraction of the first home range (i.e., the first argument) that is intersected by the second home range (second argument). Hence, changing the order of arguments will lead to a different result (see *Fieberg & Kochanny, 2005* and vignette("hr-overlap") for more examples).

```
hr_overlap(kde1, mcp1)
```

```
## # A tibble: 2 x 2
##   levels overlap
##    <dbl>   <dbl>
## 1    0.5   0.506
## 2   0.95   0.628
```

For probabilistic estimators, several other home-range overlap indices have been proposed (*Fieberg & Kochanny, 2005*); these are also implemented in the amt package (again we point the reader to the vignette for hr_overlap() for more details).

## MANY INDIVIDUALS OR SAMPLING INSTANCES

Most telemetry studies collect data on several individuals and/or during several sampling instances (e.g., time periods). To compare estimates across sampling instances and to facilitate population-level inference, it is therefore important that the methods discussed so far for individual home ranges scale easily to situations with many animals and/or time

intervals. To do this, we recommend the use of *list columns*. A list column is a column of a `tibble` (or a `data.frame`) that contains a list. And a list, in turn, is a very flexible data structure that can hold almost any object. Thus, we can split an original data set by one or more grouping variables and save the tracking data of each instance in a list (i.e., we have a list of `tibbles` were each entry in the list is the `tibble` with the tracking data of one instance). We then iterate over this list to estimate home ranges (for example using the function `map` as discussed below) and store results in another list column, together with meta-information (such as the name of the individual, its sex or age, or the time period associated with when it was tracked) in other columns.

To demonstrate estimating home ranges for multiple instances (e.g., animals or time frames), we will consider the full data set containing locations from all six fishers and illustrate workflows addressing three different example questions:

1. Do estimates of home-range size differ between sexes?
2. Is there a correlation between environmental covariates and estimates of home-range size?
3. How do weekly "home ranges" change over time?

The aim of these examples is twofold: (1) we illustrate the benefit of standardized classes for home-range estimation (Fig. 1) and list columns, and (2) we highlight that some results are sensitive to the choice of estimator whereas others are not. In particular, estimates of absolute home-range size tend to vary considerably among estimators, but relative comparisons over time or space are often robust to estimator choice (*Signer et al., 2015*).

All three of the above questions require that we iterate over several animals (Questions 1 and 2), and animals and weeks (Question 3). List columns that organize data for each individual or sampling instance provide a simple way to facilitate these analyses. The function `nest()` from package `tidyr` can be used to create a list column; `nest()` only requires the name of the list column and the columns that should be nested into the list. When using the syntax `nest(data = c(x_, y_, t_, HDOP))`, below, all columns that are not named in the `nest()` call act as grouping variables. In our first example, the only column not listed is `id`, so it serves as a grouping variable; later we will show to group by more than one grouping variable. Alternatively, we could have used `nest(data = -c(id))` to specify that we want to use `id` as a grouping column, and that all other columns should be nested. These two approaches will result in identical results. Also note that we could choose a different name for the list column (i.e., it does not need to be labeled `data`).

```
dat1  <- dat %>%
  nest(data =c(x_, y_, t_, HDOP))
```

The result of `nest()` is a `tibble` with three columns: `id`, `sex` and `data`.

```
dat1
```

```
## # A tibble: 6 x 3
##   id    sex   data
##   <chr> <chr> <list>
```

```
## 1 F2     F       <track_xyt [57 x 4]>
## 2 F3     F       <track_xyt [113 x 4]>
## 3 M3     M       <track_xyt [216 x 4]>
## 4 M2     M       <track_xyt [100 x 4]>
## 5 F1     F       <track_xyt [250 x 4]>
## 6 M5     M       <track_xyt [460 x 4]>
```

data is a list column that contains a tibble with all of the relocations for a given animal (id). This list can be accessed as any other list with [[ to obtain an element from the list. For example, to obtain relocations for the first animal, we can use:

```
dat1$data[[1]]
```

```
## # A tibble: 57 x 4
##           x_        y_ t_                          HDOP
##        <dbl>     <dbl> <dttm>                     <dbl>
##  1 1780865. 2403219. 2011-01-01 00:00:30  10.5
##  2 1781973. 2402677. 2011-01-01 01:02:29  15.1
##  3 1781900. 2402526. 2011-01-01 03:00:20  30.5
##  4 1781967. 2402741. 2011-01-01 04:05:30   6.4
##  5 1781866. 2402139. 2011-01-01 05:05:01  20.5
##  6 1781289. 2402110. 2011-01-01 06:07:05  45.8
##  7 1781828. 2401963. 2011-01-01 07:06:06  19.7
##  8 1782609. 2401669. 2011-01-01 08:04:24  20.2
##  9 1782372. 2401732. 2011-01-01 09:04:07  21.2
## 10 1782237. 2401888. 2011-01-01 10:03:08  20.5
## # ... with 47 more rows
```

Next, we use the mutate() function from the dplyr package to add a new list column that contains the home-range estimate for each animal. To achieve this goal, we have to iterate over each element in the column data, apply a home-range estimator, and save the result in a list. In base R, the function lapply() is well suited to this task. An alternative is the function map() from the purrr package.

```
hr1 <- dat1 %>%
  mutate(
    hr_mcp = map(data, hr_mcp),
    hr_kde = map(data, hr_kde),
    hr_locoh = map(data, ~ hr_locoh(., n = ceiling(sqrt(nrow(.))))),
    hr_akde = map(data, ~ hr_akde(., fit_ctmm(., "auto"))),
    hr_akde_error = map(data, ~ hr_akde(., fit_ctmm(., "auto",
                                               uere = 1.67)))
  )
```

The function map() always iterates over a data structure (e.g., a vector or a list) that is provided as its first argument. The second argument to map() is a function that is to be

applied to each element of this data structure. There are three different syntaxes we may use to specify this function: (1) we can simply supply the function name, as was done for the new column hr_mcp. In this case, map() is given the tracking data of each animal (stored in the column data) and the data set for each animal is then passed to the function hr_mcp(). This syntax works because the function hr_mcp() does not require the specification of further arguments (note, the default value of 0.95 for the home-range level is used). (2) A formula (~) notation can be used to pass a function to map(). The advantage of this notation is that it is possible to access the data under evaluation—i.e., the relocation data of the current animal can be accessed either through a ., as we illustrate above, or through the predefined variable .x or ..1. For example, for the local convex hull method, we want to choose n (the number of neighbors) as the square root of the number of observations. Thus, we count the number of rows with nrow(.) and then take the square root. Similarly, for the aKDE home-range estimator, we first want to fit a continuous-time movement model to the relocation data and then use this model when estimating the home range. Thus, we first pass the data, again using the ., to the function fit_ctmm() and then pass the result to the function hr_akde(). (3) map() can be used analogously to lapply(), by passing an anonymous function. We did not use this approach here, but if we would use this for the MCP home ranges, the call would change from map(data, hr_mcp) to map(data, function(x) hr_mcp(x)). x is just a local variable (i.e., a placeholder) for the current animal's data and could also be named differently.

The data set hr1 has now gained a new list column for each home-range estimator (in total there are now five new columns).

```
str(hr1, 2)
```

```
## nested_track [6 x 8] (S3: nested_track/tbl_df/tbl/data.frame)
##  $ id          : chr [1:6] "F2" "F3" "M3" "M2" ...
##  $ sex         : chr [1:6] "F" "F" "M" "M" ...
##  $ data        :List of 6
##  $ hr_mcp      :List of 6
##  $ hr_kde      :List of 6
##  $ hr_locoh    :List of 6
##  $ hr_akde     :List of 6
##  $ hr_akde_error:List of 6
```

We now want to obtain the home-range size for each animal and each estimator using the same map-strategy. Taking advantage of the previously introduced framework, we know that a function hr_area() exists for each estimator, and that it will return the home-range size as a tibble. However, we would have to apply hr_area() separately to each column (hr_mcp to hr_akde_error). Instead, we would like to apply the function hr_area() to one list containing the home-range estimates for all of the different estimators. To accomplish this task, we need to first change from wide to long format using the function pivot_longer() from the package tidyr so that we end up with a tibble that has a column that records the estimator (MCP, KDE, LoCoH, etc.) and a second column with the estimates.

```
hr2 <- hr1  %>% select(-data)  %>%
    pivot_longer(hr_mcp:hr_akde_error, names_to = "estimator",
                values_to = "hr")
```

We first removed the tracking data (column `data` as these data are no longer needed) and then pass the resulting `tibble` to the function `pivot_longer()`. Here we need to say which columns should be turned from the wide format to the long format (`hr_mcp:hr_akde_error`). The new data set, `hr2`, will have four columns. The first two columns are `id` and `sex` from the old data set. The third column is called `estimator` (this can be controlled with the argument `names_to`) and identifies the estimator type (i.e., the old column names). The fourth column is called `hr` and contains the actual home-range estimates (the name for this column be controlled again with the argument `values_to`).

```
str(hr2, 2)
```

```
## tibble [30 x 4] (S3: tbl_df/tbl/data.frame)
##  $ id       : chr [1:30] "F2" "F2" "F2" "F2" ...
##  $ sex      : chr [1:30] "F" "F" "F" "F" ...
##  $ estimator: chr [1:30] "hr_mcp" "hr_kde" "hr_locoh" "hr_akde" ...
##  $ hr       :List of 30
```

The new long data format allows us to apply the `hr_area` function to each element of the new column `hr` of `hr2`.

```
hr2.area <- hr2 %>%
    mutate(hr_area = map(hr, hr_area)) %>%
    unnest(cols =hr_area)
```

We now undo the list column (with the function `unnest()`). This step is necessary to obtain a `tibble` without a list column, that is suitable for plotting.

```
head(hr2.area, 2)
```

```
## # A tibble: 2 x 7
##   id    sex   estimator hr       level what         area
##   <chr> <chr> <chr>     <list>   <dbl> <chr>       <dbl>
## 1 F2    F     hr_mcp    <mcp>     0.95 estimate  4273285.
## 2 F2    F     hr_kde    <kde [7]> 0.95 estimate 10568491.
```

We can now visually explore differences in home-range size between males and females, and consider how these differences are influenced by our choice of home-range estimator (Fig. 3A; full code to reproduce the plot is given here: https://zenodo.org/record/3991482). Absolute estimates of home-range size differ considerably across the five estimators (Fig. 3A). An advantage of aKDE is that individual confidence intervals are available, and we can explicitly account for telemetry error (see *Fleming et al., 2020* for a discussion). Next, we averaged estimates of log home-range size over individuals to get population-level estimates and calculated t-based confidence intervals (with $n-1$ degrees of freedom; Fig.

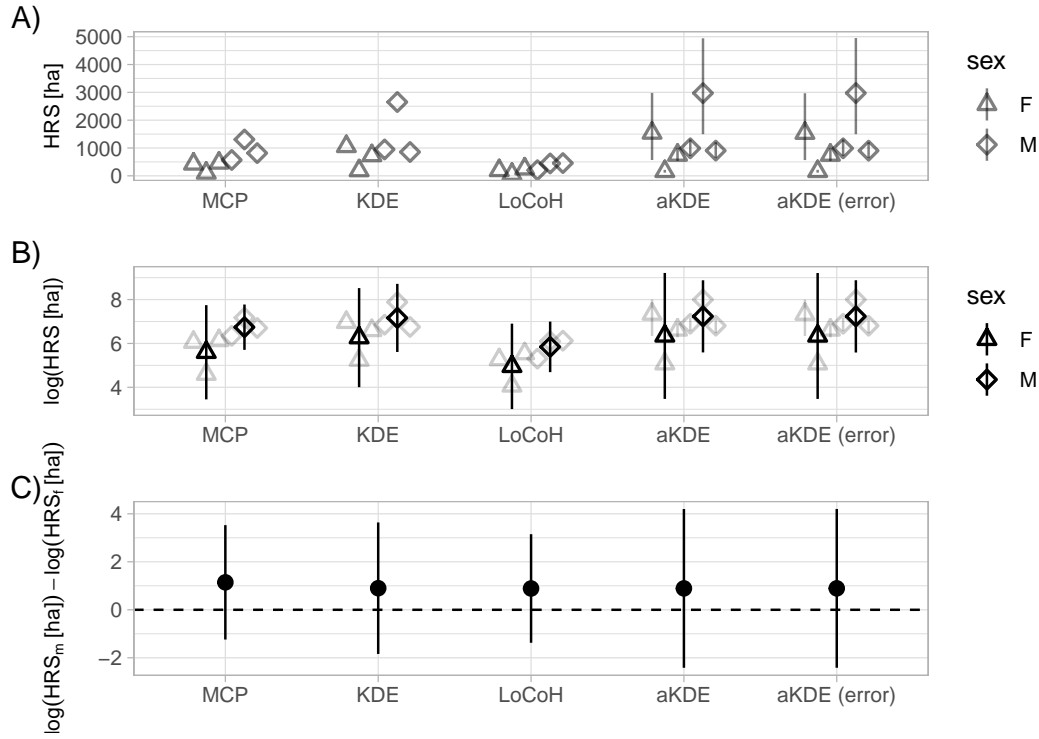

**Figure 3** **Sexual dimorphism in the size of fisher home ranges.** Different estimators (*x*-axis) lead to very different home-ranges sizes (HRS; A). The individual sex is indicated by the symbol (triangles for male and circles for female). An advantage of autocorrelated Kernel Density Estimates (aKDE) is that they provide individual confidence intervals (small vertical lines in A and B). Population- level estimates were obtained by averaging individual estimates of natural log home-range area (B) and calculating t-based confidence intervals for the mean (B). Finally, we calculated the difference between male (m) and female (f) mean log home-range size (C). The horizontal dashed line at 0, indicates no difference between mean male and female home-range size. For panels B and C we calculated 95% t-based confidence intervals.

3B). Differences between estimators becomes negligible and lead to the same biological conclusion (i.e., we were unable to detect a difference between mean log home-range size of male and female fisher; Fig. 3C). If other additional animal-specific covariates were available and of interest, we could use a linear (mixed) model to quantify the relative importance of different covariates in determining home-range size. A larger sample size (i.e., more animals) and a standardized collection scheme would be desirable (*Börger et al., 2006a*; *Börger et al., 2006b*).

In a second example, we explore whether home-range size correlates with the amount of forest within an animal's home range. To address this question, we load a preprocessed land use raster (see https://zenodo.org/record/3991482 for full details) and assign it to the object env. As before, we make use of pivot_longer() to obtain one list column with all home-range estimates and then obtain the isopleth levels with hr_isopleth() function. Again, this works for all implemented estimators in the package amt and results in an sf-object. With the function extract() from the raster package (*Hijmans, 2020*), the pixel values within each of the home-range isopleths can be queried.

```
env <- raster("data/forest.tif")
hr1.env <- hr1  %>% select(-data)  %>%
   pivot_longer(hr_mcp:hr_akde_error, names_to = "estimator",
                values_to = "hr")  %>%
   mutate(forest = map(hr, ~ raster::extract(env, hr_isopleths(.))))
```

In a final step for this analysis, we calculate the proportion of forest pixels within each individual's estimated home range and the home-range size. For the proportion, we iterate again over the list column `forest`, but this time we use the function `map_dbl()`, a variant of `map()` that will always return a numeric vector.

```
hr1.env1 <- hr1.env %>% mutate(
   prop_forest = map_dbl(forest,~ mean(unlist(.))),
   area = map(hr, hr_area)) %>%
   select(id, estimator, prop_forest, area) %>%
   unnest(cols = area)
```

We need to make a call to the function `unlist()` within the function `mean()` because `extract()` returns a list, allowing for more than one polygon per feature as is common with some home-range estimators (e.g., LoCoH). For this application, however, we can safely combine the land cover classes for different polygons belonging to the same animal. We use the resulting `tibble` `hr1.env1` to plot estimates of home-range size against the proportion of the estimated home-range composed of forest (Fig. 4). Similar to the previous example (Fig. 3), we observe that different home-range methods result in vastly different estimates in absolute terms, but the observed pattern (i.e., home-ranges with a higher proportion of forest tend to be larger in size) is consistent among all estimators. In situations with more tracked animals and more (environmental) covariates, linear (mixed) models could be used to simultaneously explore multiple determinants of home-range size (*Börger et al., 2006a*; *Börger et al., 2006b*).

Lastly, we consider an example exploring the extent to which individual space-use patterns change over time. To do so, we first add a new column to the `tibble` with the week of the year (called week) and then group our data set by animal id (`id`) and the week (week). This results in a new `tibble` where the relocations for each animal and week are stored in the column `data`. The data were sparse for some weeks, and thus, we only considered weeks with at least 10 observations.

```
hr2 <-dat %>%
   mutate(week = floor_date(t_, "week")) %>%
   nest(data = -c(id, week))  %>%
   mutate(n = map_int(data, nrow)) %>%
   filter(n > 10)
```

We then follow the same design pattern as before, combining all estimates into one list column in the long format and applying the function `hr_area` to all estimates.

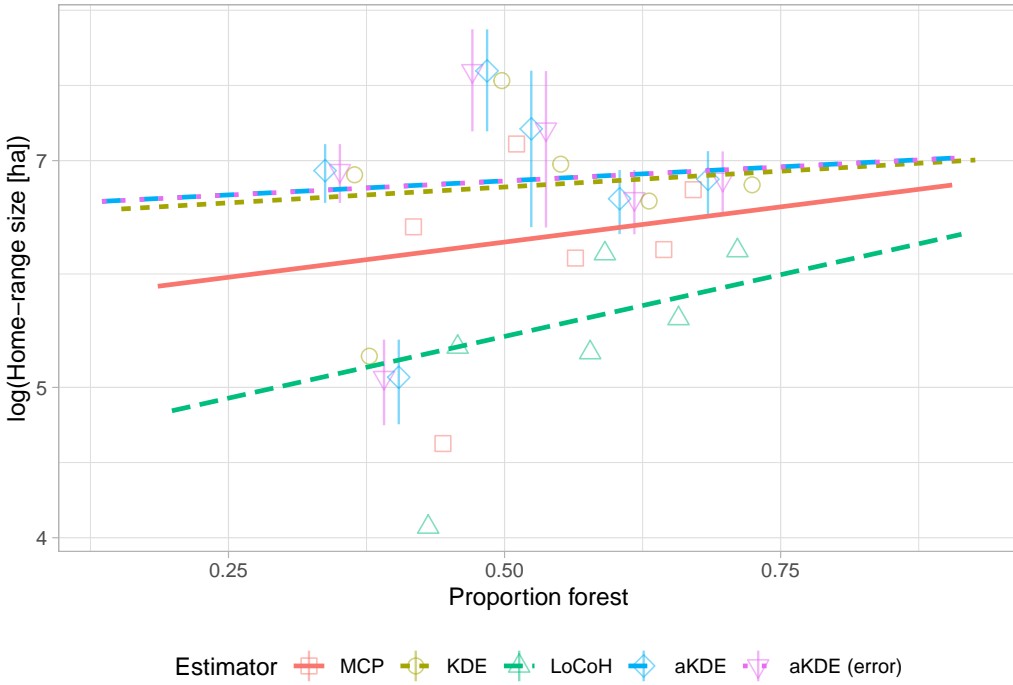

**Figure 4** **Natural log of home-range size versus the proportion of forest within the home range with linear trend lines.** Different estimators (symbols and line types) lead to very different absolute home-range sizes ($y$-intercepts) but very similar trends (slope of the regression lines). For autocorrelated kernel density estimates (aKDE) with and without error model, 95% confidence intervals are shown (vertical lines).

```r
hr2 <- hr2 %>%
  mutate(
    hr_mcp = map(data, hr_mcp),
    hr_kde = map(data, hr_kde),
    hr_locoh = map(data, ~ hr_locoh(., n = ceiling(sqrt(nrow(.))))),
    hr_akde = map(data, ~ hr_akde(., fit_ctmm(., "auto"))),
    hr_akde_error = map(data, ~ hr_akde(., fit_ctmm(., "auto",
                                          uere = 1.67)))
  )
```

As with the other two examples, there are large consistent differences between the five estimators, but all exhibit similar trends over time (Fig. 5). See https://zenodo.org/record/3991482 for the full code.

## DISCUSSION

The home range is an important biological concept that has and will continue to be highly influential. *Fieberg & Börger (2012)* argued we should clearly distinguish the biological concept of a home range from the statistical methods used to gain insights into this concept. It is also important to recognize that home-range estimates are not always the end
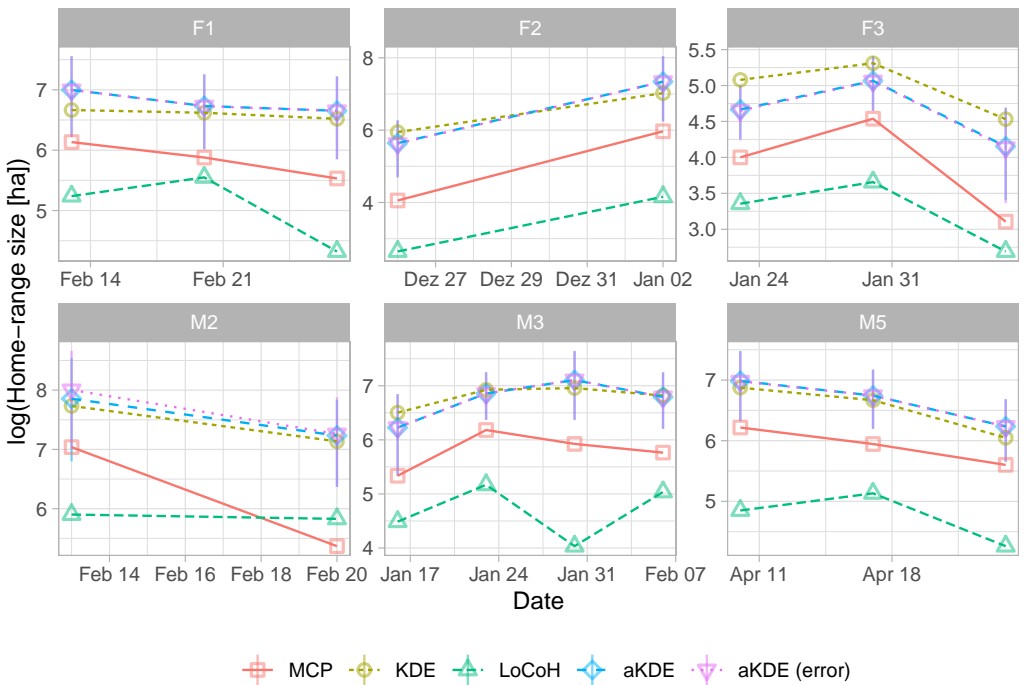

**Figure 5 Natural log of home-range size for different fisher estimated using five different estimators (line type and symbols).** For autocorrelated kernel density estimates (aKDE) with and without error model, 95% confidence intervals are shown (vertical black lines). Each panel is for one fisher (the first letter of the ID gives its sex). Individuals were followed for different time periods.

goal. Rather estimates of home-range size are often used to explore questions regarding how various factors influence animals' use of space (*Börger et al., 2006a*; *Börger et al., 2006b*). Often, these questions involve comparisons of home-range estimates over space or time and for different population segments. For example, researchers may correlate estimates of home-range size with demographic traits, landscape features that also vary in time, or the presence or absence of predator species (*Ditmer et al., 2018*; *Tingley et al., 2014*; *Van Beest et al., 2011*). Estimates of animal home-ranges are also often used to determine habitat availability when studying habitat selection. Although we agree with *Fleming et al. (2016)* and *Horne et al. (2020)* that *range* and *occurrence* distributions are useful estimation targets that can help end users choose an appropriate statistical home-range method, for some situations it may not be clear which of the two concepts (if either) is most suitable for addressing a particular research question. For example, a range distribution may not always be appropriate for studying temporarily varying space-use patterns. On the other hand, researchers may want to incorporate areas that were likely known and accessible to the animal but not used during a specific observation window when studying habitat selection (i.e., they may be interested in more than just the animal's movement path, which would be estimated by the occurrence distribution).

Analysts often seek simple measures to detect changes in spatial extent of movements over time, between sexes or habitats. Like *Signer et al. (2015)*, we found that answers to

questions that involve relative comparisons of home-range size were robust to estimator choice. Yet, differences in the tracking regime (VHF or GPS) and sampling rate (i.e., how often is an animal tracked) can lead to vastly different home-range estimates depending on one's choice of estimator (*Noonan et al., 2019*; *Peris et al., 2020*). These differences can also influence estimates of derived quantities and observed relationships, for example scaling laws between home-range size and body size (e.g., *Noonan et al., 2020*). Thus, it is important for researchers to conduct sensitivity analyses to determine how the choice of estimator influences their quantitative and qualitative results. The standards we outline here should make this task simple to accomplish.

In the spirit of allowing users to freely apply and evaluate multiple estimators, we suggest approaching home-range estimation in a consistent manner, using a standardized workflow that facilitates quantification of space use for different subsets of available data (e.g., formed by unique individuals, possibly further grouped by different temporal units such as months, seasons, or years). Animal tracking is still in its early stages (*Kays et al., 2015*), standards for tracking data are still in flux (*Campbell et al., 2016*), and new methods are constantly being developed, which will hopefully fit in the suggested framework. As animal tracking studies enter the "Big Data" realm (e.g., *Tucker et al., 2018*), a standardized, scriptable and reproducible workflow is needed to ensure reproducibility of results (*Archmiller et al., 2020*; *Lewis, VanderWal & Fifield, 2018*), and a standardized implementation of home-range estimators should help facilitate that vision.

## ACKNOWLEDGEMENTS

We are grateful to comments on earlier version of the manuscript from the associate editor, Scott LaPoint, Chris Fleming, Michael Noonan and an anonymous reviewer. We acknowledge support by the Open Access Publication Funds of Göttingen University.

### Funding

John R. Fieberg received salary support from the Minnesota Agricultural Experimental Station and the McKnight Foundation. There was no additional external funding received for this study. The funders had no role in study design, data collection and analysis, decision to publish, or preparation of the manuscript.

### Grant Disclosures

The following grant information was disclosed by the authors:
Minnesota Agricultural Experimental Station and the McKnight Foundation.

### Competing Interests

The authors declare there are no competing interests.

## Author Contributions

- Johannes Signer conceived and designed the experiments, performed the experiments, analyzed the data, prepared figures and/or tables, authored or reviewed drafts of the paper, and approved the final draft.
- John R. Fieberg conceived and designed the experiments, analyzed the data, prepared figures and/or tables, authored or reviewed drafts of the paper, and approved the final draft.

## Data Availability

Data and scripts are available at Zenodo:

Johannes Signer. (2020, August 19). A fresh look at an old concept: Home-range estimation in a tidy world. Zenodo. http://doi.org/10.5281/zenodo.3991482.

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
