# Peer review of "A fresh look at an old concept: home-range estimation in a tidy world"

_PeerJ, doi:10.7717/peerj.11031_

## Round 0.1 · original submission · Minor Revisions

We were fortunate to have three strong reviews of your manuscript. All the reviewers agreed that yours was a substantial contribution to the study of home ranges, but all three also had a number of suggestions for improvement. I found the manuscript generally clear and well written, though I am not qualified to comment on the computational aspects. In reading the manuscript, I came across a number of typos and places where I thought word choice or sentence structure could be improved. I have provided an annotated pdf in which I highlighted potential problems and used inserted comments to suggest alternative wording or explain the problem. I used this pdf also to comment on a small number of minor changes proposed by a reviewer with which I did not agree.

Editor Comments
L57 It is not clear what you mean by ‘agnostic’ in this context. Can you find an alterntive expression that would be clearer to readers?
Fig. 1 Please define all abbreviations in the caption. Even though they are defined in the text, the figure should be more able to stand alone.
Fig. 2. It appears to me that the points are gray, not black.
Fig. 3, 4, and 5. I agree with Reviewer 3 that these figures could cause problems for anyone who reproduces the manuscript as a black-and-white photocopy or who is color blind. I suggest using different symbol shapes and line types to differentiate categories. These should also be defined in the caption, even when there is a legend/key.
Fig. 5. The caption should identify the abbreviations for males and females, mention the log 10 scaling of the home range size, and note the differences in time of year and duration of study for different individuals. For M3, please use regular numbers rather than scientific notation for the y-axis to match the other panels.

·

Basic reporting

The manuscript describes new computational tools that enable researchers to apply a range of space-use estimators to animal tracking data. The ultimate goal of the work appears to be to provide researchers with a single platform for estimating these quantities. The ability to apply a range of different home range estimators in R, using a clean and straightforward workflow is of benefit to the broader community. Unifying multiple analytical frameworks is not an easy task, so I applaud the authors in their effort. The authors use clear and unambiguous language, have referenced the appropriate literature, have shared the raw data in an accessible way, and the work is self-contained.

Experimental design

The authors have designed a workflow that allows researchers to straightforwardly estimate quantities related to animal space use from animal tracking data. Experimental design, in its traditional meaning, is not relevant to the present study. Nonetheless, I do have reservations about the way several of the tools function.

First, the methods for estimating AKDE home range areas appear to require users chose an autocorrelation model as an option of the function (P. 11). Identifying the correct autocorrelation model for a specific dataset is one of the most important steps of AKDE home range estimation, and is a step that can be incredibly challenging in practice (note: there are currently dozens of unique models that can be selected from). This is especially true if users do not have access to any diagnostic tools to check that models’ assumptions are not being violated and that the fit is reasonable for the data. Given that the goal of the present work is to provide users with a streamlined workflow for estimating home range areas, I would recommend that the authors update the workflow of their package to be based on ctmm.select() by default instead of ctmm.fit(). The former performs automatic AICc based model selection, removing the subjectivity of the model selection step. I am not opposed to providing users the option of comparing AKDE estimates from different autocorrelation models as an additional option, but I would not make manual model selection the default.

Second, the authors briefly describe how users can estimate home range overlap, but no information is provided on what overlap estimator is implemented. For instance, the methods available to estimate overlap for probabilistic home range estimates are different from those that can be applied to geometric estimates, which can result in varying degrees of accuracy; Fieberg & Kochanny 2005). While the authors recognize this, it is unclear to me how the hr_overlap() function in amt accounts for this. The text on page 8 of the manuscript appears to say that it is a measure of intersection for geometric estimators, but that it can be based on one of “several other home-range overlap indices” for probabilistic home range estimates. Reading the helpfile on the hr_overlap() function from the version of the amt package currently available on CRAN, I can see no option for selecting which overlap measure is applied. Some more information on how this function works is needed if users are to know what the overlap numbers represent.

Third, all of the AKDE based home range estimates include confidence intervals on the area estimates. Confidence intervals on point estimates are particularly useful when applying an inferential framework, such as the male-female comparison presented in Fig. 3. Nonetheless, these are not depicted in the figure, and appear not to have been propagated into the comparison of male vs. female home range sizes. As a minimum, I would recommend that the authors include the CIs in Fig. 3. However, I would also encourage the authors to employ meta-analysis methods for these estimates using either the metafor R package, or the meta() function in the ctmm package.

Validity of the findings

All underlying data has been provided, the code is publicly available and well commented, and the findings based on sound science.

Additional comments

On P. 19, the authors mention briefly how the currently framework makes it easier to compare home range estimates via sensitivity analyses. This can be an incredibly challenging step in practice for many ecologists. While I recognize that these types of sensitivity analyses were beyond the scope of the present work, some guidance on how to approach this would likely benefit readers. For instance, what specific tools might research apply (e.g., cross-validation, bootstrapping, etc.), are there any functions in the amt package for doing this?

·

Basic reporting

The authors demonstrate a convenient software package to perform home-range analysis across multiple individuals and multiple estimators, which the authors also promote, as a kind of sensitivity analysis. The topic is timely and the software offers clean solutions.

Experimental design

* With Scott LaPoint's fisher data, there is a complication that worries me in that the location errors are pretty big (10-100 meters) compared to how far the fishers travel during the sampling period (~10 minutes). There is error information in that data (e-obs inaccuracy estimate in meters) and the calibration was calculated in https://www.biorxiv.org/content/10.1101/2020.06.12.130195v1 S4.3.1 to be roughly 1.67. So when working this these data in ctmm, I would normally do something like

uere(FISHER) <- 1.67
GUESS <- ctmm.guess(FISHER,CTMM=ctmm(error=TRUE))
FIT <- ctmm.select(FISHER,GUESS)

to fit with calibrated error in the model, which will return a less tortuous model with smaller home range. The occurrence distribution will also become less tortuous and less constrained to the location point estimates. I would check to make sure that this doesn't have a large impact on the results, though I recognize that this data complication is a big tangent to the purpose of this paper.

There is a second complication with the fisher data, specific to movement model fitting, in that some of the individuals have accelerometer triggered sampling, so there is only sampling when the movement is more tortuous (which biases the fitted movement model). But I don't have any solutions for that.

Validity of the findings

> For convenience, and as is common in the literature, we will refer to the suite of methods used in this context as home-range estimators, even though these methods may have different statistical estimation targets (Horne et al. 2020).

I know this is the tradition, but it is also the tradition that there's no difference where there is, and when one class of estimators does a much better job at estimating Burt's notion of the "home range", don't you think it's confusing to researchers if both classes of estimators are referred to as "home-range estimators". Is there a better term that could be given to this, like "home-range and travel-log distributions" or something?

* I think there is a danger in using the occurrence distributions like this in Figure 5, as the confidence areas are quantifying a combination of animal movement and sampling quality. If the location fixes start to fail or become more erroneous, then the OD's area will change, even if the movement behavior is not changing.

* The IID OD in Figure 5 (I'm assuming no error model) is just the empirical distribution and so its area calculated here is probably just an artifact of the pixel resolution. Because the trends are all so similar, that then makes me wonder if the dominant effect here isn't the number of successful fixes. Can you check that?

> For example, a range distribution will not be appropriate for studying temporarily varying space-use patterns.

If p(x,y|t) is a function of time (non-stationary), then it can make sense to estimate p(x,y|t) as approximately constant over different windows of time as an alternative to modeling the time dependence explicitly, like performing a moving average versus a fitting a trend model. In either case, p(x,y|t) is something to be estimated and should not to be conflated with a time-average of p(x,y|t,data), which would be an occurrence distribution.

I think the danger of estimating range distributions over narrow windows of time is that the effective sample size can be too small to reliably support that: https://besjournals.onlinelibrary.wiley.com/doi/10.1111/2041-210X.13270

* If it was up to me, I would change the x-axis of Figure 4 to be the proportion of forest within the occurrence distribution, even though the y-axis uses the range distribution. In my mind, such a figure would encapsulate the most appropriate usage of both distributions, as using the range distribution on the x-axis can give false positives, while using the occurrence distribution on the y-axis would give negative bias.

Then, having an example with the occurrence distribution, Figure 5 could be dropped. But if keeping Figure 5, I would switch from occurrence to range distributions, while making sure that the time window can support that. For fishers, that might limit you to bi-weekly or weekly home-range estimates. I would also include the CIs where possible.

Additional comments

* Is there any chance the confidence intervals could be included on the AKDE point esitmates in Figures 3-4? I know the other estimators don't produce confidence intervals on their point estimates, but ideally they would/should.

* Can you explain the t-based confidence intervals in Figure 3? The mean area parameter estimate sampling distributions are given t-distributions? Where do the 3 parameters of that distribution come from?

Reviewer 3 ·

Basic reporting

Some minor problems with references.

Background could benefit from reorganizing.

Raw data is shared.

See below for more comments.

Experimental design

The gap is clearly defined in this manuscript: a common and reproducible workflow for estimating animal home ranges.

The code examples for a single individual and for multiple individuals show how this can be performed using amt.

See below for more comments.

Validity of the findings

no comment

Additional comments

In general, this paper provides a thorough set of examples for estimating animal home ranges with the amt package. The figures included (Fig 3, 4, 5) compare the results from different home range estimation methods available in amt. I appreciate the authors’ recommendation of sensitivity analyses to determine the influence of home range estimation method on results. However, I believe the authors’ have not sufficiently accompanied the amt’s functions with introductory information around the dichotomies of home range estimation methods mentioned (eg. “old” and “new”, geometric and probabilistic, and range distribution and occurrence distribution). In addition, the meaning and repeated emphasis on the authors’ implementation being “tidy” and “coherent” is lost for the reader, especially to one without a knowledge of the various packages and communities of the R programming language. I recommend that authors bolster the introductory information with clear comparisons of different home range estimation methods, potentially including those not currently available in the amt package, to assist the reader in selection and evaluation of home range estimators. In addition, a simplified description of the proposed home range estimation framework, using amt and other package dependencies, will help communicate the methods to a broader audience. Please see specific comments below.


General comments

I think this paper has the opportunity to clearly describe and categorize home range estimators. Since the goal of this paper is to present users with a standard workflow and to help users evaluate the influence of home range estimator selection, it should be clear how the available methods are fundamentally different. Here are some of the descriptors used throughout the paper:

L29: “statistical and modelling” - Are these different? Which of the different approaches presented fit into these categories?

L33-34: “range distribution” and “occurrence distribution”

L39: “old estimators”. These are introduced after the
mention of advances/new estimators. Consider reorganizing (see below).

L72: This is another dichotomy: geometry and probabilistic, but it wasn’t mentioned in the introduction.


Though this information is not clearly presented at the moment, I believe it is mostly contained within the paper and simply needs reorganizing to better present it. The introduction could more linearly track the historical progression from early uses (eg Darwin L25) to “old” estimators (L39) and advances “addressing issues related to autocorrelation” (L37), finally describing the niche amt is filling (L39:L52, L62:L70). This could also provide an opportunity to expand on the autocorrelated nature of GPS data and how that is being addressed by new methods.

In the current version, the “conceptual framework” section begins with a contrast between two types of home range estimators, geometric and probabilistic estimators, before moving quickly into the structure of the amt package’s functions for estimating home ranges. This structure is arguably no longer a “conceptual framework” and rather an introduction to the functions and structure of amt. Perhaps this section could be reorganized to include a comparison of estimation methods using the descriptors mentioned throughout the paper, before moving onto a description of amt.

Following this, I believe the reader would greatly benefit from a more detailed, sequential description of the home range estimation methods available in amt. What methods are available in amt and what are their parameters/arguments/methods? How are these methods characterized using the aforementioned descriptors? What are the key references for each method? Each function could have its own distinct paragraph followed by global methods for all types.

For example, in the current version the description of the local convex hull method and the expectation to provide the square root of the number of observations is mixed in with a description of the map function (L255:L257).


The following comments pertain specifically to the precision of the sections regarding amt’s functions, package dependencies and the language used to describe outputs and variables. I believe it is critical when writing about programming to use precise and simple language to avoid confusing the reader. In addition, all examples should be presented at a minimum reproducible level, for example there is no need to discuss approaches that are tangential, or include tools that are not necessary.

L105-107: Great! This is a really clear example of how amt makes it easy to work with different home range estimators.

L108: Replace “a tibble is very similar to a data.frame” with “a tibble is an extension of a data.frame” and please describe specifically how it is different. “improved properties” does not explain to a user why it is being employed over the base R data.frame. According to this (https://r4ds.had.co.nz/tibbles.html#tibbles-vs.-data.frame), the differences between tibbles and data.frames have only to do with subsetting, printing, and some default options when creating a tibble or data.frame. If I am not missing something, please remove “A tibble makes it is easy to work with list-columns” because this is not something that distinguishes tibbles from data.frames. To clarify: I hope that the authors’ will emphasize why they are using, for example, list columns rather than emphasizing a specific package’s function which is not different than base R.

L110, L156: The function hr_isopleth does not return a tibble. It returns an object which is of class sf and data.frame. A reader might benefit from a few clarifications: what is an sf object/what is the sf package? What is the meaning of the column sfc_POLYGON?

L119-120: The Supplement 1 points to a Zenodo repository (https://zenodo.org/record/3991482). In the repository the “all-code.R” file does not contain any preparation steps for the dataset “fisher_preprocessed.rds” that are not presented in the manuscript.

L122-124: Is it necessary to load the tidyverse package? I believe that it is clearer for the reader if you only use specifically the packages that you need, as opposed to metapackages such as the tidyverse. In this section “One individual or sampling instance”, you do not present any figures so there is no need to mention ggplot2. In addition, there is no manipulation of dates so there is no need to mention lubridate. On the other hand, where does the “filter” function come from? Where does the “CRS” function come from?

L131: I would try and use language consistent with R help manuals here. Perhaps “Results from any estimator in amt are stored in a named list with three elements: crs, data, levels.” Then describe each in the subsequent sentences, dropping the semicolon.

L190: Instead of framing this in terms of what amt lacks, it would be clearer for the reader if you simply stated that to estimate home ranges of multiple animals you recommend using list columns. On this note though, list columns are not unique to tibbles. They are simply a column type that can be used with data.frames, data.tables, etc. This could be simplified by specifically stating why list columns are used and how, when combined with the map function, they can be used to estimate home ranges for multiple animals.

L377-384 (also L1, L19): What is specifically “tidy” about the proposed workflow? I understand that there are a subset of R packages released under the “tidyverse”. Does “tidy” therefore mean that the workflow is compatible or perhaps dependent on these tidyverse packages? It is not clear to me why the authors use this word often, other than it being a trending term in a subset of the R community that does not seem to carry an obvious meaning in this case.

L377-384: Similarly, I’m not sure anyone but a reader or user can decide if this is a “coherent” workflow. Perhaps a different word choice? I believe this is an impressive “standardized” workflow, that allows users to easily estimate home ranges by different methods.

Figure 1: This figure and its caption uses confusing language: hr_locoh has “type” and “n” - these are parameters for the function. In contract, hr has “levels”, “crs” and “data” - these are attributes of the resulting object. In addition, why show the empty boxes when there are no specific methods or attributes?


Minor comments

L13: Please clarify or consider rewording “computational point of view”.

L32-33: Please replace “whether they estimate one of two estimation targets:” with “whether they estimate either”.

L37: Could you expand on the autocorrelated nature of GPS data and, briefly, how these new methods address this issue?

L38: Please remove “modern day”.

L62: Please use more confident language: “we propose a general framework that can accommodate most home-range estimators”.

L69: Please review the citation style for the reference “Wickham and others 2014”.

L69: Why is the R Core Team cited in reference to the amt package? Maybe it could be clarified with something like this: “using the package amt (Signer 2019) for the R programming language (R Core Team 2020) and a previously published…”

L70: Please include species name for “fishers”

L97: Please replace “and/or” with “and”.

L98: Please remove “for example”.

L99: This paper describes amt. Please remove “here and in amt” and simply refer to the name of the property in amt.

L100: Please remove “should also”.

L117: Consider citing a reference for Movebank. According to their website (https://www.movebank.org/cms/movebank-content/about-movebank#citing_movebank) you may cite either or both listed in the references below.

L126: What is the meaning of the “init:epsg:5070” string?

L128: What is the meaning of the “default reference bandwidth”?

L130: Please remove “note just the ones illustrated here”.

L196: Please change “to demonstrate list columns” to “to demonstrate estimating home ranges for multiple animals” or something of the sort.

L203: Please use “absolute home range size” as in Fig 4 caption.

L203, L369: These two sentences repeat the same information. This is an example of where information presenting differences in home range estimation methods is not clearly organized for the reader.

L227: Please consider replacing vague and assuming “in the regular way” with a description of subsetting with square brackets.

L242: Please explicitly state which package the mutate function comes from and possibly what it does.

L244, L260: Why bother mention lapply if you don’t use it? This could be confusing for a reader.

L375: Please add “the” - “how choice of estimator”.

L377-380: Please consider simplifying this sentence, its meaning is unclear.

L377-384: I think this paragraph would be better off restating how the amt package and the proposed sensitivity analyses allow a user to use and compare different home range estimator methods using a common tool. Does the amt package provide or intend to provide all “new methods” that are being developed to maintain this common interface for users? That could be mentioned here if so. I would also argue that with papers such as Tucker et al. 2018, wildlife biology is definitely already in the era of big data and your proposed home range estimator functions are a welcome tool for engaging with it.

L397: Please correct this citation from “Beest, Floris M van” to “van Beest, Floris M”.

Figure 3, Figure 4, Figure 5: Colors selected are not colorblind safe or greyscale friendly. Please consider the viridis color palette or other colorblind safe palettes.

Figure 5: Please correct “ggplot” to “ggplot2”. The function geom_smooth uses a smoothing method, which one is selected for this plot? Please mention it in the caption. Please replace “year day” with “Day of Year” or “Julian Day”. Consider changing the break points in the M5 panel because 82.5 is an impossible value for integer type Day of Year. Consider mentioning the different x-axis scales, presenting from 4 days to >30 days.


Comments on Supplemental material

The following comments pertain to the “all-code.R” file included in the Zenodo archive (https://zenodo.org/record/3991482/files/all-code.R).

Please remove knitr chunk labels (presumably returned by knitr::purl) and replace with informative comments and perhaps sections matching the paper’s section names. As described above, hr_area returns a tibble, but hr_isopleth does not.

In addition, when I ran the all-code.R file, I encountered a number of messages, warnings:

• messages: “ff is invalid” many times and “Minimum sampling interval of X minutes in unknown” many times
• warnings: >50 warnings mentioning the CRS object and 25 warnings mentioning the geom_smooth function


References

Kranstauber B, Cameron A, Weinzierl R, Fountain T, Tilak S, Wikelski M, Kays R (2011) The Movebank data model for animal tracking. Environmental Modelling & Software 26(6): 834–835. https://doi.org/10.1016/j.envsoft.2010.12.005

tibbles vs data.frames in R for Data Science. https://r4ds.had.co.nz/tibbles.html#tibbles-vs.-data.frame

Tucker et al. 2018 https://science.sciencemag.org/content/359/6374/466.abstract

Wikelski M, Davidson SC, Kays R [year]. Movebank: archive, analysis and sharing of animal movement data. Hosted by the Max Planck Institute of Animal Behavior. www.movebank.org, accessed on [date].

---

## Round 0.2 · Minor Revisions

I apologize for the delay in completing my decision on your revised manuscript. I had other time-limited responsibilities that caused the delay.

All three reviewers were available to read the revised manuscript, and all recommended acceptance. However, all reviewers also provided minor suggestions to improve the manuscript. Unfortunately, I cannot relegate these to the production process. Therefore, my decision is in the category of ‘minor revisions’. After you have seen the suggestions and decided your best course, I will accept the revised manuscript without sending it out to reviewers, so it shouldn’t take much longer. I have a couple of comments on the reviewers’ suggestions.

Reviewer 2 suggested adding color to the last two figures. Since you made the figure black-and-white in response to my request to make it accessible to a color-blind reader or to half-tone photocopies, I will accept them as they are. However, if you wish, you could add color while retaining the different shapes and lines.

Reviewer 3 asked you to be consistent in your use of ‘home range’ and ‘home-range’. However, I find that you are consistent in using the hyphen when the context is adjectival and no hyphen when home range is a noun. This is grammatically correct, so no changes are needed.

Reviewer 1 suggested using the same scale (linear vs. log) to facilitate comparisons in panels A and B in Figure 3. I had assumed that you used a different scale because the range of values differed. Indeed, values in A range up to about 3,000 and values in B range up to about the 10 to the eighth power. However, the caption implies that the data in B should be means of those in A. I must be missing something here, but other readers might also be confused.

·

Basic reporting

no comment

Experimental design

no comment

Validity of the findings

no comment

Additional comments

After carefully reading the revised manuscript, I have found that the authors have adequately addressed all of my initial concerns in their revisions. I only have one minor comment that I leave up to the authors' discretion. In Figure 3, panels A and B contain similar information, but A is on a linear scale, and B on a log. I would recommend putting these on the same scale as this would allow for these to be more readily compared.

·

Basic reporting

no comment

Experimental design

no comment

Validity of the findings

no comment

Additional comments

I appreciate all of the work that the authors have put into the revisions, even though some of my recommendations turned out to be unnecessary worrying. I still think "For example, a range distribution will not be appropriate for studying temporarily varying space-use patterns." is too strong of a statement, and now seems in conflict with Fig. 5 as I understand it, but the authors are allowed to have their own opinions. My only additional comment is that the last two figures might be easier to decipher with some color.

Reviewer 3 ·

Basic reporting

no comment

Experimental design

no comment

Validity of the findings

no comment

Additional comments

This manuscript provides a clear set of examples for using the amt package
in calculating and comparing home range estimates. The authors have
made significant improvements through the previous round of revisions. My
comments here are few and minor.

Throughout the manuscript, "home-range" and "home range" are both used. Please consider selecting one and using it throughout.

Lines 114-124: This paragraph, describing the output formats of the hr_area and hr_isopleth functions, serves an important role for describing the outputs of these functions.
Two object types are presented: tibbles and sf objects.

"the function hr area() in the package amt, will always return a tibble"
"Similarly, the function hr isopleth() in amt will always return a data.frame with a list column where the isopleths are saved as simple features as implemented in the sf package"

I am not sure what is similar in these two: hr_area returns a tibble, and hr_isopleth returns a "data.frame with a list column...". Please remove "similarly". In addition, I am unsure if "will always" is necessary in these cases, unless the authors of the `amt` package are planning on introducing backwards-incompatible changes.

I appreciate the clarification presented here on why tibbles are used instead of
data.frames. I think, however, the description of sf objects could be improved (Lines 120-124, 178-180).
While sf objects are in fact a data.frame, I think "return a data.frame with a list column where the isopleths are saved as simple features as implemented in the sf package" is overly wordy and could be rephrased simply as "return an sf object". This is immediately
clear to any user that is familiar with sf objects. Then the following sentence
could be used to clarify further for those unfamiliar. For example, taken from the sf package's README: an sf object "represents simple features as records in a data.frame or tibble with a geometry list-column".

R: class(hr_isopleths(hr_mcp(deer)))
[1] "sf" "data.frame"

Note as well, much of the information presented in this paragraph is repeated
throughout the rest of the manuscript. While I appreciate this repetition and
reinforcement helps a user understand how to use the `amt` package for home-range
analysis, some more justification type repetition could be pared down ("makes it easy to perform further GIS-type work").

hr_area: Lines 114-120 & Lines 164-167
hr_isopleth: Lines 123-124 & Lines 178-180

Lines 142-145: Comments concerning the definition of coordinate reference systems.
I appreciate the clarification of setting the CRS using EPSG codes. However,
given recent updates to GDAL and PROJ, R spatial has changed their expectations
with regard to setting coordinate reference systems. This is the relevant
page describing this: https://www.r-spatial.org/r/2020/03/17/wkt.html.
This presents itself in two places relevant to this manuscript.

Firstly, running the lines:

R: library(amt)
R: data(deer)
R: hr_isopleths(hr_mcp(deer))

Returns the following warning:

Warning message:
In CPL_crs_from_input(x) :
GDAL Message 1: +init=epsg:XXXX syntax is deprecated. It might return a CRS with a non-EPSG compliant axis order.
Calls: hr_isopleths ... st_crs<- -> st_crs<-.sfc -> make_crs -> CPL_crs_from_input

Highlighting internal code in `amt` that should be updated to use the new syntax.
In addition, in the manuscript (Lines 142-146) the CRS is set using the following:

CRS("+init=epsg:5070"))

My interpretation of the above link is that the recommended way to set a CRS
in R for use with the `sf` package is

st_crs(5070)


Finally, I think that writing a manuscript describing code, analytical details
and trying to reach an audience of mixed programming skill can be challenging.
There are innumerable ways of coding any one task, and describing each of these
approaches can be overwhelming. In the case of this manuscript, there are
a few instances where the authors present their method then
in parenthesis or as an aside, mention an alternative. I believe that
for a reader, especially a novice, presenting only your approach can
yield a more succinct and clear manuscript. This might help avoid reader
confusion (eg. map is used to iterate but in base R, lapply could work too).
This is simply a comment, not meant for any substantial changes. Something to
consider when parts of the manuscript feel bogged down in details or confusing.

---

## Round 0.3 · accepted · Accept

The authors have responded appropriately to the previous minor comments. I recommend acceptance.